# The Possibility of Including Donkey Meat and Milk in the Food Chain: A Southern African Scenario

**DOI:** 10.3390/ani12091073

**Published:** 2022-04-21

**Authors:** Zahra Mohammed Hassan, Tlou Grace Manyelo, Ndivho Nemukondeni, Amenda Nthabiseng Sebola, Letlhogonolo Selaledi, Monnye Mabelebele

**Affiliations:** 1Department of Agriculture and Animal Health, College of Agriculture and Environmental Sciences, University of South Africa, Florida 1710, South Africa; zahrabattal@gmail.com (Z.M.H.); nemukn@unisa.ac.za (N.N.); sebolan@unisa.ac.za (A.N.S.); letlhogonolo.selaledi@up.ac.za (L.S.); 2Department of Agricultural Economics and Animal Production, University of Limpopo, Sovenga 0727, South Africa; manyelo.t.g@gmail.com

**Keywords:** health benefit, milk, meat, functional food, donkey

## Abstract

**Simple Summary:**

The demand for donkey products such as meat and milk has increased in recent years, as new information on the health benefits of the products is becoming more available. Recent nutritional trends have shown a preference for nutritional and functional foods, giving consumers more options to choose from. Donkey products are seen as valuable foods that contain appreciable nutraceutical properties. However, less information is available on their optimal management practices, and their level of introduction to the food chain remains minimal. Increasing awareness of the importance of donkey products will increase their productivity and accelerate the introduction of these products into the food chain. This review aims to document available information on donkey products and factors affecting their introduction into the food chain, quoting different scenarios from the Southern African region.

**Abstract:**

Animal proteins are essential for the optimal growth and health of humans. Meat and milk are common sources of protein, mostly produced by ruminants. The agrarian challenges experienced around the world warrant sourcing alternative proteins from animals that can withstand harsh environmental conditions to produce quality proteins. Donkeys (*Equus asinus*) are known to survive on low husbandry and inferior quality forage to produce meat and milk, which have been used since ancient times. However, the commercialisation of these products has not flourished due to product scarcity, low production rates, and consumer preferences. Recent discoveries and the quest to look for alternative sources of protein have sparked studies on donkey products. In addition, milk and meat from donkeys have active ingredients that could also contribute to curing diseases. Donkey milk is believed to contain antioxidant, antimicrobial, antiproliferative, and antidiabetic properties. In many countries, particularly in Africa, the consumption of donkey meat and milk has not been fully adopted due to a lack of knowledge and legislation regarding production.

## 1. Introduction

Access to diverse and healthy food ingredients has increased in recent years. For researchers to take part in revealing diverse food resources for human consumption, different potential food resources need to be identified and evaluated. Donkeys (*Equus asinus*) are domesticated animals that belong to the equine family [1]. They have been used as working animals, mainly in packs, since ancient times. They are believed to have originated in north-east Africa and remain an important work animal in difficult mountainous landscapes. Blench [2] states that donkeys are not traditionally perceived as a source of meat, but rather seen as assistant animals for transport, carrying goods, and ploughing. However, in recent years, interest in donkey products has increased due to new research showing the importance of these products for nutrition and health. Mature male donkeys are slaughtered for their carcasses and females are reared for their milk [3,4]. Consumer preferences and food norms have played negative roles in the use of donkey products. The preference for some animal products is usually influenced by factors such as culture, religious beliefs, or economic status. Milk is an important part of human nutrition in providing essential nutrients such as water, carbohydrates, fats, proteins, minerals, and vitamins. Due to different allergies to cow’s milk, alternative sources are being sought. Donkey milk has a higher concentration of minerals than human milk, but significantly lower than ruminant milk [5]. It has been utilised for a long time [6]. It is believed that an old practice in the Maasai community in Kenya is to feed warm donkey milk to children suffering from diseases such as severe coughs, pneumonia, and common colds [7]. Recent trends show donkey milk being commercialised in some countries and used as a better alternative for infants and people who are allergic to milk protein from other sources [8]. Polidori et al. [9] report that donkey milk is safer and more similar to human milk in comparison to milk from other mammals. Due to its good palatability and tolerability, it is being recommended by paediatricians as an alternative to infant milk [10].

Donkey milk is also used in cosmetics, as it is believed to nourish the skin and maintain its radiant look. It was mentioned that the famous queen Cleopatra used to immerse herself in donkey milk to preserve the splendor of her skin [11]. Donkey milk is rich in vitamins and polyunsaturated fatty acids [12] and contains antiaging, antioxidant, and regenerating compounds, which are described as being naturally active in skin hydration and skin aging prevention. The lipids in donkey milk contain essential fatty acids and saturated fatty acids. Furthermore, it is believed that the antibacterial properties of compounds such as lysozyme and lactoferrin can inhibit the growth of pathogenic bacteria on skin and decrease skin infections. Lactoperoxidase (LPO), which is an oxidoreductase enzyme, has protective ability against microorganism infections. It can catalyse the oxidation of subtrates, which has bactericidal activity against Mycoplasmas and monocytogenes [12].

Different studies report that donkey milk has antioxidant, antimicrobial, antiproliferative, anti-inflammatory, and antiviral properties [13,14,15]. Although donkey milk has recently come into favour because of its nutrient content, it is not produced in large quantities. Although milk from donkeys is the most prevalent product, donkey meat has gained some interest in recent years as valuable food for human consumption due to its quality proteins [16]. Donkey meat is characterised by low fat and cholesterol contents and it is rich in iron [17]. This is of interest as more people desire lean and more exotic foods [18]. Despite the recent knowledge of the significance of donkey meat and milk, the use of these products is less popular in comparison to those of ruminant animals such as cattle and sheep. This review aims to document available information on donkey products and factors affecting their introduction into the food chain, quoting different scenarios from the Southern African region.

## 2. Common Donkey Breeds in Southern Africa

Donkeys are believed to originate from the wild asses of Africa and Asia, with the African continent having two known breeds, namely the Nubian wild ass from the north and the Somali wild ass from the east [19]. Donkeys population is declining in many countries due to economic development, urbanization, and increased demand for donkey hides [20]. Articles about donkeys no longer appeared in farming journals [21]. Their early history is difficult to reconstruct due to the scarcity of archaeological data. However, all African wild asses may be a single population with variations by geographic area. Somali wild asses may be the only surviving African wild asses; they may be a sister group to domestic donkeys [22]. Donkeys were probably domesticated in north-east Africa [23] and emerged in most semi-arid parts of Africa, with their distribution spreading towards the eastern and southern parts of Africa [2]. Other studies report that the African ancestors of the current species of the donkey were the Nubian wild ass and an unknown ancestor that is different from the Somali wild ass [24,25]. The Somali ass (*Equus*
*asinus somaliensis*) is believed to be the surviving African wild ass and is close to the domesticated donkey.

Ancient Portuguese documentation on the Zambezi Valley in 1758 notes the presence of donkeys in Zimbabwe. It was also thought that these donkeys might have been brought into north-east Zimbabwe along the gold routes from Mozambique in Sofala [26]. However, it has been suggested that donkeys in Zimbabwe were spread by the Boer farmers into the inland from North Africa [27]. Nengomasha et al. [28] report a lack of information on the types of donkey breeds available in Zimbabwe because its typical traits have not been established.

In South Africa, donkeys are reported to have arrived in 1650 with settlers from Europe and were used as pack and draft animals on farms [29]. It is also believed that the movement of the settlers into southern and central Namibia in the second half of the 18th century led to the introduction of donkeys to the country [30]. Damaras, Namas, and Tswanas are known to be large consumers of donkey meat. Among the Ovambos, Hereros, and Kavangos, the consumption of donkey meat is very low [30]. The donkey in South Africa is small, with colours ranging between black, dun, brown, and light grey. These donkeys can be traced to the northern parts of Africa and are used mainly for ploughing and cart pulling for the transportation of firewood, water and building materials, as well as for farm cultivation. The average body weight of the donkeys found in Botswana and Zimbabwe is 140 kg [31,32].This is considered relatively small. Donkeys have also been used for milk production in the south-western part of Botswana due to the recommended health benefits [33]. The majority of the donkey population in Africa is found in Ethiopia, with the rest being scattered in other regions [34,35].

Gifford-Gonzalez [36] and Mitchell [37] suggest that the reason for the lack of expansion of donkeys into southern Africa in pre-colonial times was the spread of infectious diseases within sub-Saharan Africa among both cattle and dogs. However, due to recent development initiatives, the number of donkeys is increasing in some southern African countries such as Malawi and Zambia [38].

## 3. Donkey Population and Farming Systems in Southern Africa

The demand for *ejiao*, a traditional Chinese medicine that has some health benefits, has contributed to the rapidly increasing demand for donkey hides. This will have an effect on the global donkey population, donkey welfare, and livelihoods of people that rely on donkeys [39]. Therefore, efforts to formalise the donkey farming system have been proposed in South Africa’s North-West province [40]. In 2017, the North-West provincial legislature sent a delegation to China to negotiate donkey business trade between the province and the Chinese government. The intention is to establish a trade partnership that will facilitate the export of donkey hides, meat and milk, which could help accelerate rural development in the North-West province [36]. The donkey population in Southern Africa varies by country (Table 1). According to Norris et al. [41], the 2018 donkey population in Zimbabwe was estimated to be 585,048, in Namibia it was 153,126, and in Mozambique 49,428. In South Africa, although donkeys have played vital roles in transportation and tilling, the modernisation of agricultural systems and mining has led to the neglect of the role of donkeys and consequently resulted in a decline in the donkey population [7]. Nonetheless, small-scale farmers prefer donkeys to technology in farming for reasons such as affordability, ease of management, and longevity [25]. In Lesotho, donkeys are culled and their meat is used when they lose strength as working animals [21]. This practice may be responsible for the reputation of donkey meat being tough and more suitable for salted meat-based products [42]. From 2013 onwards, there seems to have been a gradual decline in the population of donkeys in some Southern African countries. In South Africa, the decline in the donkey population has led to their exclusion from education and training programmes, as they are no longer considered livestock [18]. In Botswana, the decrease in the donkey population is a concerning issue requiring urgent intervention, as it reached an all-time low of 140,000 donkeys in 2017 [43] (Table 1).

Farming systems can affect the composition of donkey milk. An extensive farming system of dairy donkeys can alter the fatty acid composition and increase the fat-soluble vitamin concentration and fat content [47]. Modern farming systems promote sustainability and good quality yields without harming the environment. This section embraces the requirements for forage and water provision, appropriate bedding, adequate space, and appropriate group size, which are believed to play crucial roles in respecting the best practices for the feeding of donkeys [48]. Therefore, systems such as diversified farming are promoted because they aim to integrate economic and ecological benefits for sustainable agriculture. It is important to understand the donkey farming system so that diseases and production parameters can be monitored. A study conducted by Daddy et al. [49] reported that 11.5% of donkeys tested in South Africa’s North-West province in the Ngaka Modiri Molema district municipality were positive for leptospira antibodies. Leptospirosis is one of the zoonotic diseases that affect donkeys, and developing strategies to monitor and treat affected herds could be very helpful to the donkey farming community. Management of intensive donkey farming system parameters such as the breeding life, mortality, and reproductive performance using the latest technologies is lacking [39]. In the Southern African Development Community (SADC), there is no formal donkey production system; donkeys are sourced mainly from subsistence farmers in rural communities [40]. Bennett and Pfuderer [39] propose a donkey farming system using a system dynamics model of donkey production. Key variables that could affect the donkey farming system are the size of the initial female breeding herd, reproductive performance, age of reproduction, percentage of female births, and average breeding life of donkeys.

In Malawi, donkeys rest at night in open-roofed *kholas*, which are pens with fences erected around the area. The problem is in the rainy season when the *kholas* become muddy and donkeys are forced to stand throughout the day and night. It is not surprising, therefore, to see donkeys lie down when they are supposed to be working because they are tired.

## 4. Donkey Milk and Meat

The use of donkey milk for human consumption is justified by its endowment with valuable nutrients and its tolerability. Compared to other species, donkey milk is characterised by lower fat, fat globules of a smaller diameter, and a lower quantity of saturated fatty acids [8]. The whey protein content of donkey milk was reported by Li [50] to be 49.08 g/100 g. The ratio of casein to whey protein in donkey milk and powder was 70.3:100. Table 2 shows the comparison between the compositions of donkey milk with other species.

Meat is an important component in the human diet due to its richness in valuable nutrients. The consumption of donkey meat in Africa dates back many years, but consumption of meat from other herbivores is higher. In some cultures, it is considered a taboo to eat donkey meat, which could be attributed to the fact that donkeys are always around humans. In some European countries such as Italy, France, and Belgium, donkey meat is consumed as part of the traditional diet [51].

In a study by Aganga et al. [52], donkey meat from Botswana was found to contain valuable nutrients such as crude proteins and minerals. Donkey meat is characterised by low fat, low cholesterol content, a favourable fatty acid profile, and being rich in iron [53]. It is believed to have potential as a beneficial alternative to the common red meats due to its high nutritional profile [52], having a comparatively better nutritional index [54]. However, donkey meat enters the food chain through adulteration [55]. Table 3 shows a comparison between the composition of donkey meat with other species.

In West Africa, donkeys are traded for their meat when they are old, sick, or exhausted. Because of its ambiguous status, the trade in donkeys for meat remains poorly documented. Reports of donkey meat being used in the food chain make it seem as though donkey meat is dangerous for human consumption. This results in a wide lack of acceptance of donkey meat in the food chain. In Kenya, most of the licences offered to legalise slaughterhouses are export-only licences, mostly because of the taboo around the consumption of donkey meat locally [56].

## 5. Functional Properties of Donkey Milk

The low microbial content and rare presence of pathogenic bacteria in donkey milk are ascribed to the natural antimicrobial substances, mainly lysozyme and lactoferrin, present in donkey milk. Donkey milk has antibacterial properties against bacteria such as *Escherichia coli*, *Salmonella enteritidis*, *Listeria monocytogenes*, *Staphylococcus aureus*, *Bacillus cereus*, *Enterococcus faecalis* and *Shigella dysenteria* [59]. In addition, donkey milk has high lysozyme activity (1402.50 u/mL of milk) and high levels of other active ingredients such as bioactive peptides, n3 fatty acid, casein, and lactoferrin [60]. Lysozyme is an important enzyme that has shown antimicrobial activity against some pathogenic bacteria, being able to inhibit viruses, fungi, and parasites. It is known for its ability to hydrolyse polysaccharides of bacterial cell walls, preventing the development of bacteria [59,61]. It also has the ability to induce cells to produce nitric oxide, a vasodilator that helps improve blood flow and reduce blood pressure [62]. Donkeys above the age of 15 years, which are at their early lactation stage, are reported to have the highest lysozyme activity values [59].

A study conducted by Massouras et al. [60] reported that all milk samples from healthy Cypriot and Arcadian Greek donkeys exhibited antimicrobial activity against *Listeria monocytogenes* and *St. haemolyticus*. Furthermore, Yvon et al. [62] report that donkey milk has anti-inflammatory properties by restoring the endogenous levels of antimicrobial peptides which are produced by paneth cells; therefore, the consumption of donkey milk could lead to a restoration of the functionality of paneth cells. The immuno-stimulating ability and anti-inflammatory effects of donkey milk are useful in treating some immune-related diseases in humans and preventing atherosclerosis [63]. Furthermore, calcium is considered a suitable nutraceutical for geriatrics, and its ample content in donkey milk supports its use in geriatric nutrition [64].

Donkey milk shows quite a low total bacterial plate count, the recorded mean value range is 2.40–5.87 log CFU/mL. This could be attributed to the good health of donkeys, the size and anatomical position of the udder, as well as the presence of natural antimicrobial components such as lactoperoxidase, lactoferrin, immunoglobulins, and lysozyme [65]. This highlights the importance of hygiene in donkey management and the lack of it, which might affect the quality of donkey milk. A study by Cavallarin et al. [4] reported high *Pseudomonas* spp. counts in raw donkey milk, suggesting possible contamination due to the use of contaminated water and poor cleaning of milking machines and other dairy equipment. This finding highlights the need to improve hygiene practices during milking and milk storage on donkey dairy farms.

However, other studies have reported that donkey milk is mostly composed of Gram-negative bacteria, which raises questions regarding the contributions of antimicrobial agents that have been reported in the literature [66]. Therefore, it would be worth expanding the study of donkey milk in relation to its antimicrobial properties, metagenomics, proteomics, and metabolomics. These studies will shed more light on the antimicrobial activity of donkey milk [66].

Furthermore, donkey milk has been utilised for its medicinal and cosmetics properties for a long time, leading it to be referred to as a “pharmafood”; it has been used to cure whooping cough and arthritis. Donkey milk is high in lysozyme, which can act against pathogenic microorganisms. In addition, the contents of calcium and liposoluble vitamins make it a favourable nutraceutical food. The active antimicrobial substances such as lysozyme and lactoferrin are also found in other mammals, but are lower in quantity (Table 4).

## 6. Factors Affecting Donkey Milk Production

Donkey milk is known to be an alternative milk source for consumers, especially infants and the elderly population [72]. Valle et al. [47] report that the extensive farming system of dairy donkeys can alter the fatty acid composition and increase the fat-soluble vitamin concentration and fat content. Furthermore, production and quality are affected by several factors such as season, the number of milkings, stage of lactation, nutrition, and breed [66,73].

### 6.1. Lactating Season

Several studies report that the season can have an effect on the production and quality of donkey milk. According to Cosentino et al. [74], Aspri et al. [12], and Martini et al. [73], during the cold season donkeys exhibit high or increased milk production compared to during warm seasons. Moreover, Polidori et al. [9] reported increased milk production in donkeys lactating in winter than those lactating in summer. However, Ragona et al. [75] observed high milk yields in Amiata donkeys when lactating during warm seasons. Martini et al. [73] attributed the seasonal differences to the different latitudes of the farming areas and the responses to the environmental conditions linked to the origin of the donkey breeds they studied.

### 6.2. Number and Times of Milkings

The number of milkings has been reported to affect donkey milk. Salimei et al. [76] found that donkeys milked twice per day (morning and afternoon) had high milk yields. Others reported that morning milking produced a lower milk yield than afternoon milking. D’Alessandro and Martemucci [77] reported high milk yields in donkeys milked three times per day in their study investigating the effects of daily milking number and frequency on donkey milk production. Moreover, Alabiso et al. [78] report high milk production when donkeys were milked three times per day compared to twice a day. Polidori et al. [9] observed increased milk production in donkeys milked three times per day every three hours than in those milked six times per day every three hours, which did not increase milk production. Muhatai et al. [79] observed that donkeys produced more milk from morning (1.1 kg/donkey) than afternoon (0.9 kg/donkey) and evening milkings (0.9 kg/donkey), when the donkeys were milked at 08:00, 13:30, and 19:30 respectively. They stated that the difference could be attributed to the longer lapse period prior to the morning milking, which is approximately 9.5 h, compared to the lunchtime (5.5 h) and evening milkings, at 5.5 h and 6 h, respectively. According to Salimei and Fantuz [66], data on donkey milk yields are expressed in mL per milking session, which is from 3 h before mechanical milking until exit from the milking parlour.

Furthermore, a study by D’Alessandro and Martemucci [77] suggested that milk synthesis processes are influenced by circadian rhythms, with the greatest secretory activity occurring overnight. The increase in milk yield per milking induced by the longer milking interval of 8 h compared with 3 h corresponded to a decrease in the secretory ability of the mammary gland [80]. Their study showed that the mammary gland can continue producing milk during an extended milking interval of 8 h, likely due to the ability of the cistern to dilate, leading to a compensatory effect with respect to the decrease in milk synthesis.

### 6.3. Stage of Lactation

Several studies report that increased milk production in lactating donkeys occurs in the first three months of lactation. Federica et al. [81] observed high milk production in Ragusana donkeys during the first month of lactation. Giosuè et al. [82] and Bordonaro et al. [83] reported decreased milk production in Ragusana jennies between 8 and 10 months of lactation. Salimei and Fantuz [66] observed decreased milk production in Martina Franca donkeys from day 30 to the fourth month of lactation. Salimei and Chiofalo [84] and Giosuè et al. [82] also observed a decrease in milk production in lactating donkeys in the ninth month of lactation. D’Alessandro and Martemucci [80] revealed that the peak of lactation was reached at day 48, followed by a slow decline, in a study conducted on Martina Franca jennies.

### 6.4. Feeding

Lactating donkeys require dietary supplementation during months when feed is not in abundance. Donkeys have low but constant daily production during lactation [8]. For high milk yields, donkeys require a diet with balanced energy, crude fibre, vitamin, mineral, and protein contents. It is important to ensure that lactating donkeys have free access to fresh feed and water for optimal milk production [85]. Feed intake is reported as low during early days of lactation, but increases when lactation increases, as does milk production. According to Pearson et al. [86], donkeys are normally fed crop residues and bush grasses, which have already matured and are less nutritious. Feed intake quantities vary with weight, milk production, weight gain, work intensity, and stage of gestation or lactation [87]. It is estimated that the energy requirement per kg of milk produced by a jenny is about 2.54 MJ of net energy (NE) throughout the whole lactation period [87]. Donkeys take advantage of their ability to recycle large amounts of nitrogen, and their energy and protein requirements are much lower when compared to other equids [87]. In donkeys, the consumption of legume hays is higher than grass hays and straw [87]. Martin-Rosset [87] recommended the use of variable amounts of cereal-based concentrates, according to the nutrient value of the forage, in order to balance the energy and protein intakes. Furthermore, based on adaptated indications provided by Martin-Rosset [87], a lactating donkey ought to ingest 3.3 kg DM/100 kg of BW as forage and 1.65 kg DM/100 kg of BW as concentrate during the first three months of lactation [48].

Donkeys subjected to high- or low-energy diets with similar body weights showed different performances, whereby donkeys fed a high-energy diet had high average daily gain and feed efficiency levels compared with those that received a low-energy diet [88]. As Polidori and Vencinzetti [89] reports, pregnant or lactating donkeys should be provided with ad libitum access to hay or haylage during the last trimester of pregnancy and the first 3 months of lactation. Burden et al. [90] estimated the requirements of lactating jennies to be 2.6 kg of good hay (9 MJ DE/kg DM) + grazing and a laminitic balancer or +0.2 kg alfalfa chaff (10 MJ DE/kg DM) and a donkey- or laminitic-specific feed balancer in the first three months of lactation.

Essential vitamins such as vitamins E and A needs to be supplemented, as most donkeys have restricted access to grazing for much of the year. To ensure that balanced levels of vitamins and minerals are supplied, an equine-specific supplement or balancer may be useful, particularly in very young, very old, or sick donkeys. Growing donkeys may be particularly at risk if their diets are lacking in calcium and phosphorous. Of most common concern is a diet with an imbalance of phosphorous and calcium. Calcium/phosphorous ratios of 1:2 to 2:1 are ideal. Protein requirements for a mature donkey are estimated to be 40 g crude protein (CP)/100 kg body weight (BW) per day [91].

### 6.5. Breed

Although limited information exists on the effects of donkey breeds on milk yields, few studies have mentioned the suitability of some breeds for donkey milk production. For example, Ragusano breeds are listed as the most productive breeds with high milks yield containing high levels of lactose [56,58,92]. This breed has been reported to produce average milk yields of 1.64 ± 0.79 kg per day, with an average body weight range of 300 to 350 kg [76,77]. The Amiata donkey breed has been reported to produce a milk yield of 735.0 ± 83.5 mL/milking with an average body weight of 311 ± 43 kg [82]. Salimei et al. [85] reported a milk yield of 1 469.2 ± 77.6 mL/day in Martina Franca with a 93 kg average body weight. Ivanković et al. [92] reported a milk yield in Croatian Littoral Dinaric donkeys of 317.83 mL/milking and of 745.38 mL/milking in Istrian donkeys with an average body weight of 218 kg. According to D’Alessandro and Martemucci [78], differences in milk yield are attributed to the type of breed and other factors, such as breeding techniques. This should direct the farmers to choose the right breed as far as high milk yield is concerned.

### 6.6. Metabolic Factors

Metabolic disturbances such as Dyslipidemias occur secondarily to physiological status (such as pregnancy, lactation, and food deprivation) and pathological status (such as stress, gastrointestinal disease, endotoxemia, respiratory problems, liver diseases, parasitism, and laminitis) [93]. These processes are linked to a negative energy balance in donkeys. Animals in negative energy balance experience several physiological and metabolic changes, which may incline them to several negative effects such as poor reproduction performance and poor immunity [93]. The state of negative energy balance (NEB) leads to economic losses through decreased milk production and decreased reproductive performance [94].

## 7. Constraints of Introducing Donkey Products into the Food Chain

The donkey population in Africa is threatened by the interest in donkey hides, which are used in different industries [58,95]. The demand for the hides is not proportionate to the demand for other products such as meat, resulting in wastage of the carcass. This could be offset by educating people on the importance of donkey meat for human consumption. Although both horse milk and donkey milk have been reported as other alternative sources, their contribution to the global milk production remains low, estimated to be below 0.1% [96,97]. Several factors contribute to the delay in introducing donkey products into the food chain. A few of these factors are discussed below.

### 7.1. Consumer Perception

In some Southern African countries such as South Africa, cultural beliefs and food taboos are followed strictly in some communities, which in turn influence their food consumption and consequently their health [98]. Most of the taboos are passed down from previous generations, but some knowledge is transferred among different social groups [98]. Food taboo practices are the outcome of religious and secular factors [99]. The preference of consumers for other sources of milk and meat is considered one of the factors deterring the introduction of donkey products into the food chain. Although there is some acceptance of donkey products in countries such as France, Italy, and other parts of Europe, its introduction to the food chain in Africa will take some time. Since what consumers want can change with new facts and scientific evidence, it is worth continuing to conduct research on the neutraceutical properties of donkey products. Atabek [100] revealed that food products are very sensitive to consumer perceptions, particularly dairy products. Furthermore, Yegbemey et al. [101] and Keitshweditse [102] state that older people tend to have a positive perception of the consumption of donkey milk, which could be attributed to health concerns that older consumers might have. Tackling the negative impacts of food norms is crucial, because they are considered some of the important drivers of food security [103,104].

### 7.2. Low Production

Donkeys are known for their low production of milk, which can hamper supply into the food chain. They provide low but constant daily production [9]. The stage of lactation, foaling season, and milking technique are some of the factors that affect milk production [80]. One jenny produces about 4 cups (1 litre) of milk per day. Thus, the milk is very difficult to find and considered a rare item [14]. This normally results in an increase in the price of donkey products, particularly milk. In Africa, donkey milk is believed to have less economic importance [25]. Donkeys are not given the same attention as other animal species, which could be a contributing factor to the scarcity of donkey products. Veterinarians are rarely called to attend to the donkeys by the owners, unless the services are provided for free [105].

### 7.3. Lack of Welfare Protocol

Despite the valuable contributions of donkeys to the human population, little information is known about their proper care and management [106]. Donkeys have played important roles in the livelihood of poor people; they are used for farming and transportation and the milk is used for infants to compensate for human milk. It is a valuable substitute for infants suffering from multiple allergies and also useful in cosmetic production. It can also be considered the best “pharmafood” for people experiencing distinctive nourishment hypersensitivities and skin and bone issues. Donkeys in developing countries are believed to endure harsher conditions than in industrialised countries where they are exposed to little work and have access to good nutrients [106]. It has been reported by Raspa et al. [48] that there is generally a lack of comprehensive assessments of dairy donkey welfare. The best protocols for donkey welfare are not available [44]. The appropriate nutrition welfare criteria have been assessed only through the evaluation of body condition scores [107]. Slaughtering donkeys mainly for their skins also leads to neglect of the welfare of the animals, as this practice places little value on the meat. This helps propagate the greater profit that can be obtained from buying sick or weak donkeys at a lower price, as the skin will be worth the same as that from a more expensive, healthier animal. Furthermore, farm management requirements such as forage and water provisions, appropriate bedding, adequate space, and appropriate group size are also important [107]. Donkeys can survive and have a working life of up to 15 years if management is optimal [108]. In addition, the recent interest in trading in donkey skins to produce products such as *ejiao* has led to inhumane treatment and slaughtering of donkeys [109].

### 7.4. Nutritional Factors

Donkeys are non-ruminant herbivores with a large intestine that works as the primary site of microbial activity [110], which occurs in the cecum and colon. Information on the nutritional requirements is conflicting. For instance, the daily dry matter intake of donkeys is estimated to be 1.75–2.25% of body weight [111], but Fielding and Krause [111] estimate it to be 2.5–3% of body weight.

Although donkeys are highly efficient at digesting fibre of poor nutritional quality, the provision of small amounts of concentrate feed (0.3–0.5 kg per animal) would probably have a more beneficial effect than supplementary fodder. The estimated protein intake for a mature donkey is about 40 g crude protein (CP)/100 kg body weight (BW) per day [112]. The energy requirements are about 14.4 and 17.1 MJ DE/day in summer and winter, respectively [113]. However, whether this is a viable option for poor farmers in developing countries is questionable. For owners to spend extra resources on nutrition, it has to be justifiable; in this case, that would mean returns in the form of milk and meat. Contrary to this, Burden et al. [114] state that donkeys do not require energy-rich cereal grains or products high in molasses, which are often associated with the development of health issues such as laminitis, gastric ulceration, hyperlipaemia, and colic. The conflicting information on the nutritional requirements for donkeys could present a challenge when identifying the optimal options. The study by Aganga et al. [115] illustrates that donkeys are supposed to have access to supplementary feed when they are expected to perform extra work or when the grazing range has been depleted. However, the donkey owners and handlers normally do not show an interest in the consumption intake capacity of the donkeys, leading to poor feeding welfare.

### 7.5. Complex Legislation

Food system certifications offer several advantages, including increasing the value of the certain products or of the process through which they are obtained, in turn guaranteeing the sensory and nutritional characteristics of the products and their quality and uniqueness [3]. In some countries, although legislation for the consumption of donkey products is clear, this has not prevented the incorrect use of donkey products. For instance, in Kenya the consumption of donkey meat has been legal since 1999 [116], with the aim of curbing backyard slaughter, improving food safety, and stimulating donkey production in response to market availability. Nonetheless, donkeys are being slaughtered inhumanely and the meat is sold fraudulently as beef; this practice is believed to have been fuelled by the demand for donkey skin [117,118]. A few countries such as Burkina Faso banned the export of donkey meat and skin to Asia after realising that the demand is unsustainable [118]. South Africa legally exports about 10,500 donkey hides to Hong Kong and mainland China on an annual basis; however, the real quantity is believed to be higher, as smuggled donkey hides are frequently reported [119]. In Botswana, although donkey meat is considered a valuable product, donkeys are slaughtered mostly for their hides. This is done in most cases illegally and inhumanely, and as such the Botswana government decided to ban the export of donkey products [119].

In sub-Saharan communities, eating donkey meat or donkey meat products is still seen as taboo, although a few tribes in countries such as Kenya, Botswana, and South Africa are known to consume donkey end products as affordable sources of protein. In South Africa, the trade and export of donkey meat hides is legal, provided that the slaughter was performed at a registered equine abattoir [38]. According to Carder at al. [119], there is little legislation currently in place to govern the production of donkey hides and products. They further suggest that if policies and legislation were to be introduced and enforced in Africa and sub-Saharan Africa in relation to the trade, it could allow future studies to be conducted with the aim of exploring the impacts of the legislation on people’s livelihoods. However, recent information from the North-West province of South Africa is that the legislature is in the process of regulating the donkey industry in the province. Donkey product production remains controversial because of the concerns regarding animal welfare and the declining population [38].

### 7.6. Processing Difficulty

Milk products such as cheese and yoghurt are popular, but producing these products from donkey milk is not as easy as from other milk sources. Cheese made from donkey milk is not produced in the traditional way [120]. The processing of donkey products into secondary products such as cheese is considered one of the problems associated with using donkey milk. Recently, a study by Iannella [120] discovered that pure camel chymosin, an enzyme found in camel rennet, is able to clot donkey milk effectively. Although the results are promising, the products are believed to have been poorly investigated.

## 8. Recommendations

During this study, it was found that some communities, especially in Africa, have already ruled out the consumption of donkey meat and milk. There is a sense of the donkey being perceived as a friend. The introduction of the donkey as a potential protein source for humans will be met with great resistance. In some communities where there is some acceptance, this has been partly driven by the fact that new demands are being made by foreign countries, and also partly by endemic poverty, as this move is seen as a new means of survival. Incidents of donkey product consumption are often framed as scandalous acts by the media, forming negative stereotypes. Clear legislation needs to be formulated, especially by countries that have legalised trading in donkey products. Best practice guidelines on the husbandry and management of donkey products need to be established in addition to offering assistance to farmers. Governments in different Southern African countries should partner with willing non-profit organisations to empower farmers and should provide donkeys to encourage production.

Governments need to introduce legislation to curb the illegal slaughter of donkeys in order to maintain and increase the current population.

The proper breeding of donkeys needs to be funded, and work towards increasing the population needs to be encouraged. A regional body should govern and align donkey use with current opportunities in order to tap into the emerging business opportunities, and at the same time to deal with the misuse of the donkey populations in the Southern African region.

## 9. Conclusions

Donkeys are known to survive on few resources and have the ability to adapt to harsh environmental conditions. The donkey populations globally and in Southern Africa are threatened because of different emerging socio-economic drivers, mainly due to a lack of welfare and proper legislation to govern their slaughter. In recent years, donkey products have gained interest in other parts of the world as potential alternative meat and milk sources for human consumption. For the Southern African region to tap into the emerging socio-economic opportunities, appropriate measures need to be put in place to regulate the use of donkey products and increase the donkey population. Due to consumer preferences, it might take a long time for products from donkeys to find their place in the food chain in Africa. Joint efforts from farmers and animal scientists to improve donkey welfare are crucial in increasing productivity and at the same time in creating awareness of the importance of donkey products.

## Figures and Tables

**Table 1 animals-12-01073-t001:** Donkey populations in Southern Africa.

Country	Year	Donkey Population	Reference
South Africa	1996 2019	210,000 146,136	[43] [44]
Botswana	2003 2013 2017 2019	493,000 310,000 140,000 139,524	[45] [46] [43] [44]
Namibia	1996 2017 2018 2019	71,000 159,000 153,126 154,007	[43] [46] [39] [44]
Lesotho	1996 1999 2019	152,000 203,368 116,553	[43] [44] [44]
Zimbabwe	1996 2018 2019	104,000 585,048 599,780	[43] [39] [44]
Mozambique	1996 2018 2019	20,000 49,428 49,831	[43] [39] [44]
Malawai	1999 2019	2150 6376	[44] [44]
Zambia	1999 2019	1700 2200	[44] [44]
Eswatini (Swaziland)	1996	12,000	[38]

**Table 2 animals-12-01073-t002:** Nutritional compositions of milk from donkeys and other species.

Components	Donkey	Human	Cow	Goat	Sheep
Total solids (g/100 g)	8.8–11.7	11.7–12.9	12.38	13.23	15.19
Fat (g/100 g)	0.3–1.8	3.5–4.0	3.46	4.62	7.30
Lactose (g/100 g)	5.8–7.4	6.3–7.0	4.71	4.47	4.60
Totalprotein (g/100 g)	1.5–1.8	0.9–1.7	3.43	3.41	5.70
Ash (g/100 g)	0.3–0.5	0.2–0.3	0.78	0.73	0.80
Caseins (g/100 g)	0.64–1.03	0.32–0.42	0.27	0.25	0.70
Whey Proteins (g/100 g)	0.49–0.80	0.68–0.83	0.45	0.60	0.21

Sources: [57,58].

**Table 3 animals-12-01073-t003:** Nutritional composition of meat from donkey and other species.

Components	Donkey	Cow	Goat	Sheep	Pork
Moisture (%)	74.8	75.0	74.2	70.2	66.0
Fat (g/100 g)	2.02	7.9	2.8	8.1	8.2
Protein (g/100 g)	22.8	25.0	23.0	24.0	25
Cholesterol (mg/100 g)	66.7	73.1	63.8	78.2	73.2

Sources: [52,53].

**Table 4 animals-12-01073-t004:** Functional properties of donkey products.

Active Ingredient	Biological Effect	References
Lysozyme	Reductions in gastro-intestinal infections in infants, promotion of healthy growth	[66]
Lysozyme	Low bacterial concentration, immune modulation	[66]
Bioactive peptides	ACE activity, modulation of physiological functions	[65] [61] [67]
n3 fatty acid	Anti-inflammatory	[65]
Casein	Hypoallergenicity	[65]
Lactoferrin	Iron metabolism	[65]
Taurine	Enhancement of body immunity, development of cardiac muscles	[50]
Lactoperoxidase enzyme	Bacteriostatic effects against *Listeria monocytogenes*	[67] [50]
Dietary PUFAs	Maintenance of energy balance and minimisation of body fat deposition	[68] [9]
Lactobacillus plantarum	Production of bactericidal bacteriocins	[69]
α-lactalbumin	Antiproliferative activity	[70]
Lactoferrin	Preventing free radical production by the skin	[71]

## Data Availability

No data was collected as this was a review.

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
