# Peer review of "The Possibility of Including Donkey Meat and Milk in the Food Chain: A Southern African Scenario"

_animals, 2022, doi:10.3390/ani12091073_

Round 1

Reviewer 1 Report

This paper aimed to review the possibility of including donkey milk and meat in the food chain in the Southern African scenario. The topic can be of interest but the following major points should be considered.

The manuscript does not appeared to be well written. The english language should be improved greately thoughout the text. For example in line 55: “Donkey milk exhibits most minerals content than human milk” ? In line 58: ”while it is warm” ? In line 59: “common cold” ?. L 125: “farm cultivation as well as water” ? L394-396 ?  In Table 1 the sequence of years is confusing. Etc.

Technical terms must also be checked.

L71; 75-76; etc.. In my opinion, the main product of a dairy donkey farm is milk. The meaning of the term “by-product” is not clear.

L79-91. This is not clear. This manuscript is not a systematic review.

L166-168. Same statement as in L 262-263

L205-205. “donkey meat contains high biological proteins which are characterised by healthy contents of fatty acids” ?

L 375. “stomach” ?

In addition.

Paragraph 5 and 6. This is not a review on donkey milk or meat. These two paragraphs should be merged, mentioning milk for first (also focusing on milk gross composition) and then mentioning the secondary product, meat.

L276-286. This is not clear. The milking session should be mentioned. It is rather obviuos that the higher milk yield/day can be obtained by the higher number of milkings.

L306-316. This is not clear and can be misleading to the reader. There are not published studies on the effect of breed on milk yield. Again, milk yield/d obtained with different number of milkings can not be compared

Paragraph 8.4. This can be misleading to the reader. The authors should consider that there are not consolidated requirement for lactating donkeys and that nutrient requirements and suggested allowances, nowadays available specifically for donkeys, are mainly devoted to working animals, i.e., used for transportation, small agricultural works, and equine therapy, and to animals at maintenance, i.e., companion animals, often castrated, hosted in international animal rescue charities.

Paragraph 8.6. There are published informations abou the production of Yoghurt from donkey milk

References should carefully checked. Ref. 19 is the same in 42. Ref 33 is the same in 48. Ref 81 is the same in 101. Etc. Some references are listed and cited in the text with first name instead of family name of the author.

Reviewer 2 Report

Dear Authors

Please note that English is not my native language.

Thank you for the interesting paper. It contributes to the possibility of integrating donkey meat and milk into human food, ensuring food security. However, in my opinion, there are some aspects that need to be improved in order to value not only the paper itself but the knowledge and dissemination of these products.

GENERAL

At “Introduction” there should be a mention of the use of donkey milk in cosmetics.

In this paper, it would be very useful to present comparative tables with the chemical composition of donkey's meat and milk and these products from other food-producing species.

Lines 168 and following – "modern farming systems". In which they consist, specifically? In my opinion, sustainable and eco-friendly is not enough to explain the production systems of these animals.

Lines 242 and following – These total plate count values cannot be considered as characteristic of donkey milk. These values are related to factors other than the animal species. They are related to the hygienic conditions in which the milk is obtained and manipulated, the health of mammary gland, among others.

The paragraph 7 - Factors affecting donkey milk production should be re-writen:

- Terminology used should be revised. For exemple: “season” is different from breeding season or lactating season; “donkeys milked in the morning” is diferent from morning milking

- You must verify if the bibliography that you cited refers exactly what you wrote. For exemple – plaese, check line 269: the reference will be Bernabucci et al (77) or Martini et al (75)?

- You must present values that allow the reader to have an idea of the extent of the differences found. For exemple: Lines 270 and 271 – what is the average amount produced in the cold season and in the warm seson? Is it very diferent? Lines 276 to 286 – what is the amount produced with 2 or 3 milkings? How much is produced in the morning milking and in the afternoon milking? And results on the production in diferent stages of lactation?

- How do you explain the action of the different factors:

Why does extensive farming alter the composition of milk? Is it good or bad?

Why does breeding season and milking number change the amount produced?

Why does morning milking get more milk than afternoon milking?

Many of these explanations are mentioned in the bibliography you consulted. You should study it well to present these explanations.

Line 375 and following – The microbial digestion in donkeys happens in caecum and colon, not in stomach. Please, correct it

DETAILS

Line 17 – “nutreceutical”. Please correct to nutrAceutic.

Line 57 – “Salimei,.” Please, delete. The other references only have the citation number, not the name of the author

Line 99 – Gebread et al. Please, delete. The other references only have the citation number, not the name of the author

Table 1 and table 2 – these tables must be formatted:

In Table 1, we cannot understand the correspondence between the lines of the “Country” colunm and the lines of the other colunms. Why are the years random in the colunm “Year”? Example – 1996, 2019, 2003, 2013,….

In Table 2, we cannot understand the correspondence between the lines of the “Active ingredient” colunm and the lines of the other colunms.

Line 220 – “Listeria monocytogène” – please correct to Listeria monocytogenes and in italic.

Lines 221, 225 and 229 – “lysosome” – please correct to lyzozyme.

Lines 222 e 223 – “Bioactive peptides …. Casein and Lactoferrin.” Why in capital letters? Please correct.

Lines 225 and following – These two sentences can be written in only one because the antimicrobial activity of the lyzozyme results precisely from the ability in hydrolize the peptidoglican from the bacterial wall.

In table 2 – Please correct Lysosome to lyzozyme; Listeria monocytogenes to Listeria monocytogenes; There are also other microrganisms that can be inactivated by the lactoperoxidase su«ystem. Please mention some of them.

“Maintaining energy balance and minimising body fat deposition” Please correct to “Maintenance of …. Minimization…”, according to the previous text

“Prevent free radical …” please correct to “Preventing free…”, according to the previous text

Line 233 - Listeria monocytogenes to Listeria monocytogenes

Line 254 – It has already been mentioned earlier so there is no need to repeat.

Lines 262 e 263 – this statement is very vague. You must explain the reason why the extensive system can change the chemical composition etc. This statement was already wrote in lines 166 and 167. It must not be repeated although it has to be explained.

Line 295 – Feeding – it will be Feeding or farming system?

Line 357 – “… valuable contribution…” what has been this contribution? Please, specify.

Line 362 – “well fare” May be welfare?

Lines 368 and following – You should explain this sentence better, please.

Line 395 – “interest in interest in consumption”. Is it like this or a reptition?

Thank You.

Round 2

Reviewer 1 Report

The paper has been generally improved but some points still need to be considered

By-product still present in line 80.

L89-102 The paper is not a systematic review and this paragraph should be deleted. The introduction should end with the aim of the study. I suggest reporting the last statement of the simple summary “This review aims to document available information on donkey products and factors affecting their introduction into the food chain, quoting different scenarios from the Southern African region.”.

Table 2 and 3, and paragraph 5: In my opinion, tables 2 and 3 should be deleted and gross composition of milk and meat briefly summarized in the text (paragraph 5). Furthermore, some references are not appropriate. References 53, 54, 55, supporting the table on milk, are in fact on meat. Reference 16, supporting table 3 on meat is on somatic cell in milk (in addition, ref 16 was mentioned at line 79 in a statement that is rather speculative and can be removed). I suggest to quote one or two reviews on both donkey milk and meat and to remove unnecessary references. On the contrary, some results by Aganga et al (2003) on donkey meat from Botswana should be discussed in this paragraph.

The protein content and the ratio casein:whey protein in donkey milk should be mentioned. The discussion on gross composition of donkey meat should be shortened.

L214 This sentence is wrong: “donkey meat contains a high quantity of biological proteins which are characterised by healthy fatty acids.”. Please rephrase.

L238 Note that this paragraph deals only with milk.

L301-318 This paragraph is still unclear. Note that donkeys are usually separated from the foals 2 or 3 hours before milking. The “milking session” should be described (see Salimei and Fantuz 2012).

L309-311 This sentence is unclear. The number of milking should be given for both cases.

L329-334 This sentence should be moved to paragraph 7.2.

L349-355 I suggest replacing these statements mentioning what reported by Martin Rosset (2018; in paragraph “Quantities of feed ingested and feed allowances) about forages and concentrates.  

L365-377 This paragraph is still confounding, giving both results about the milk yield per milking and per day. The number of milking should be given when mentioning milk yield per day. In addition, the size of animals in different breed should be mentioned because milk yield per milking is also affected by the size (breed) of lactating donkeys.

L440-441 The main site of microbial activity in donkey is the large intestine and not the stomach. The term “stomach” in L440 should be replaced by “large intestine”.

List of References: There is still one repetition. First name instead of family name are still listed.

Reviewer 2 Report

Dear Authors

Thank you for your reply. However, there are some little things that should be corrected.

Line 17 – In your first version you wrote “nutreceutical”. The right term is nutraceutical. I wrote A in capital letters just to mark the mistake. Please correct to nutraceutical.

Lines 86 to 88 – In my opinion, this paragraph has no connection with what is mentioned above. Is it badly located? In my opinion it should not be situated at this point of the paper.

Table 1 – Please, in the year column, put the years in ascending order in all countries. In South Africa, Botswana, Lesotho, Malawi and Zambia is correct but in Namibia, Zimbabwe, Mozambique the years are disordered. It is also still missing to indicate the year in Eswatini.

Line 160 – Part of the parenthesis is missing in ref 21.

Table 2 - Please indicate the units for each parameter (%, g/100g...?)

Lines 198 to 201 – A reference is missing.

Line 245 –The values for bioactive peptides, n3 fatty acid, casein and lactoferrin are not referred in table 2. So, it is not right to quote the table on these values.

Lines 263 and following – In my opinion you should also mention that these values can be changed and increased if the hygiene rules in milking, transport and handling milk are not strictly complied with. You only refer that the milk is good because it comes from healthy and well-formed animals and due to the presence of antimicrobial componentes but if the hygienic rules are not complied, those properties are not enough to have a good milk.

Lines 291 and following – There are also metabolic reasons. Please, research the subject in the published bibliography.

Thank you

Round 3

Reviewer 1 Report

In my opinion the paper is acceptable in its present form

This manuscript is a resubmission of an earlier submission. The following is a list of the peer review reports and author responses from that submission.